# The Unappreciated Value of a Cheap, ‘Good Enough’ Method of Detecting Thyroid Cancer

**DOI:** 10.3390/jcm13237290

**Published:** 2024-11-30

**Authors:** Salvatore Sciacchitano, Massimo Rugge, Armando Bartolazzi

**Affiliations:** 1Department of Life Sciences, Health and Health Professions, Link Campus University, 00165 Rome, Italy; 2Department of Medicine DIMED Pathology and Cytopathology Unit, University of Padova, 35122 Padova, Italy; massimo.rugge@unipd.it; 3Pathology Research Laboratory, St Andrea University Hospital, 00189 Rome, Italy; armando.bartolazzi@ki.se

**Keywords:** thyroid nodules, thyroid cancer diagnosis, Galectin-3 ThyroTest, advanced molecular tests, cost-effectiveness, healthcare costs

## Abstract

The advent of advanced molecular diagnostic techniques has revealed plenty of information about signaling pathways and gene regulation in cancer, as well as new inputs for the classification of cancer subtypes, diagnosis, prognosis, and prediction of response to therapy. However, in most cases we do not have single biomarkers yet and, therefore, the final diagnosis is often rendered by the combination of multiple results by means of complex algorithms, eventually leading to an increase in their costs. The problem of the costs of such tests is particularly relevant in the case of thyroid cancer (TC), because of the observed increase in the number of patients affected by thyroid nodules (TN)s, in what is considered a global pandemic. High-income countries can afford the cost of the advanced molecular tests for such a multitude of TNs, since they are covered by private insurances. People living in upper-middle, lower-middle, and especially in low-income countries, where the costs for these advanced molecular tests are supported by general taxation and out-of-pocket payments, are exposed as a personal financial burden. Immunohistochemistry in cancer management represents an extremely cost-effective method in different clinical scenarios. In the preoperative recognition of TC, the use of such method, based on Galectin-3 and others protein markers, such as HMBE1, proved to be effective in diagnosing TC in TNs indeterminate at conventional cytology (Bethesda classification III or IV), with an extremely low cost. Moreover, Galectin-3 fulfills one of the major criteria of an ideal marker, being involved in the thyroid cell transformation. Despite this evidence, Galectin-3 ThyroTest is not considered and not even mentioned in many reviews, focused on the appropriate identification of TC, as well as in studies where the cost-effectiveness of the different approaches is comparatively evaluated. The aim of this review is to emphasize the value of the Galectin-3 based immunohistochemistry as a cheap and “good enough” method in the preoperative diagnosis of TC especially in, but not limited to, low-middle income countries.

## 1. Introduction

Cancer impacts health and quality of life. Cancer screening, early diagnosis, and appropriate treatment allow us to mitigate such impact. However, the use of the most cost-effective diagnostic tests is critical. The decision of adopting high-cost tests when other low-cost tests, with “good enough” accuracy, are available would have a negative impact on healthcare economic resources in all countries independently of their economic situation. This is particularly true in many countries where the total healthcare resources are finite and the cost of cancer care may become unaffordable, or difficult to sustain [1]. Moreover, increased expenditure does not necessarily mean better patients’ outcomes. Other factors play a relevant role too, including epidemiologic factors, structural, organizational, and cultural issues [2]. There is no doubt that early identification of cancer using screening tests, when it is still asymptomatic, allows a reduction in mortality and an increase in patients’ quality-of-life. However, the cost of both screening and diagnostic tests may exceed the economic capability, thus exposing national health budgets, insurance companies, or personal finances to an excessive burden [3]. To appropriately plan cancer prevention and management in an equitable and affordable way, there is need for an accurate priority-setting and cost-effective analysis, particularly when we consider the cost of cancer’s growing disease burden [4,5]. More than 70% of the global cancer burden, in fact, falls in middle-to-low-income settings, where most people do not have access to the resources and systems available in high-income countries [6]. A sustainable cancer screening and diagnostic program for each type of cancer should, therefore, be planned considering the local context and available healthcare financial resources and should be based on evidence derived from comparative clinical effectiveness and cost-effectiveness as well as social values of each test. This is the case for thyroid cancer (TC), whose incidence has increased in recent decades in many countries, mostly due to overdiagnosis of indolent tumors [7,8]. The economic cost associated with the management of patients with TC varies across countries and the different health systems and, in addition, it changed over time [9,10,11]. Such costs may include substantial out-of-pocket payments and indirect costs incurred by the patients [12]. It has been reported that the expenditures for thyroid cancer management is approximately three times higher in the United States than in France [9]. The selection of patients for surgery represents a critical step in thyroid cancer management. Several criteria have been applied to make the right choice, including medical history with prior exposure to RX and familial thyroid cancer, clinical examination, scintigraphy, ultrasonographic evaluation, FDG-PET, and Fine Needle Aspiration Biopsy (FNAB), which play a central role in selecting patients for surgery. However, there is still an area of uncertainty, represented by the so-called “gray zone” of indeterminate follicular thyroid tumors. In recent years, the classic diagnostic algorithm of the thyroid nodules has been integrated with advanced molecular tests, applicable to FNA-derived cytological material, in particular from those lesions which remain unclassified at conventional cytology (Bethesda categories III—atypia of undetermined significance or follicular lesions of undetermined significance and IV—suspicious for follicular neoplasm or follicular neoplasm). Current generation of molecular testing techniques proved to be accurate in identifying benign nodules that can be safely observed [13,14,15,16], thus significantly reducing unnecessary diagnostic surgery, as reported in a Canadian public healthcare setting [17]. However, this benefit comes at a significant cost. Such advanced molecular studies, in fact, are far from being cheap and their cost varies from CAD 3000 to CAD 5000 per test, depending on the specific testing procedure [18]. Advanced molecular tests have been included into the major international guidelines for the management of thyroid nodules (TN)s and TC. However, because of their cost they may not be appropriate in those healthcare scenarios, characterized by limited healthcare resources. In these conditions, patients may be exposed to inappropriate surgery, responsible for the occurrence of peri-operative or post-operative complications.

The aim of this review is to discuss the appropriate diagnostic test to be used for the characterization of indeterminate thyroid proliferations, considering the economic context and in view of the observed global increase in TC incidence that exposes patients and healthcare system to a relevant financial burden. The inclusion of a low-cost and “good enough” immunohistochemical test, based on the expression analysis of the cancer marker Galectin-3 in the decision-making strategy, instead of the expensive molecular testing, would be more cost-effective in especially, but not limited to, middle or low-income countries.

## 2. The Global Pandemic of Thyroid Nodules

TNs are frequently observed, particularly in individuals over 60 years of age. A recent metanalysis, performed to map the global epidemiology of TNs, indicated that they have an astonishingly high prevalence rate of 24.83% (95% CI 21.44–28.55). This means that a TN is found in one of every four people in the general population and indicates that TNs have become a global pandemic [19]. Such an increasing rate in the detection of TNs is rising concerns about the consequent burden that is overwhelming the finance of healthcare systems, payers, and patients [20]. In particular, the healthcare systems of low- and middle-income countries (LMICs) are experiencing difficulties in coping with the costs derived from the management of the global burden of millions of TNs detected by the widespread adoption of sensitive imaging techniques and subsequent surgical treatments, with consequent personal and global financial burden [21].

## 3. Thyroid Cancer Epidemiology and Impact on Healthcare Costs

While TNs are so frequent, TC is considered a rare occurrence. It has been reported that it can be detected in only a small percentage of thyroid nodules (10–15%) [22]. According to the American Cancer Society, approximately 44,000 new cases of TC are diagnosed each year in the US [23]. Even if until recently the rate of new TCs grew faster than for any other cancer in the US, mostly because of its increased incidental detection while performing imaging tests for other medical problems, the adoption of more stringent diagnostic criteria, was responsible for a reduction in its incidence rate by about 2% each year since 2014. Nevertheless, the death rate for TC remained unchanged since 2009 and, at least for the most common and well-differentiated types of thyroid malignancy, the survival rate is very high (up to 97%) [24]. Such increase in the incidence of an asymptomatic, non-lethal TC, that would probably not have been diagnosed had testing not been performed, has been reported in as many as 26 countries in four continents and it is often considered the consequence of an overdiagnosis [25]. The overdiagnosis of TC is increasing rapidly worldwide and has now become a major global public health challenge in both high- and, especially, in LMICs. The macroeconomic cost of 29 different cancers was recently estimated in 204 countries, territories, and for a set of World Bank regions, representing 99.7% of the world’s population and calculated as a share of gross domestic product (GDP) and per capita economic cost in 2020–2050 [26]. According to this estimate, global economic cost of TC was USD 129 billion (UI 87–193), calculated as international dollars at constant 2017 prices, with a per capital loss of 14.6 (UI 9.9–21.9). Both of these results were lower compared to what was reported in other cancers, including those affecting the trachea, bronchus, lung, colon, rectum, liver, leukemia, and breast, which accounted for the largest economic costs. However, the total macroeconomic cost for TC, was clearly higher in high-income countries (HIC)s, compared to upper-middle, lower-middle and low-income countries.

## 4. The Diagnostic Challenge of Identifying a Thyroid Cancer Among the Multitude of Thyroid Nodules

When facing a TN, the primary concern is identifying the rare malignant nodules among the multitudes of benign ones in order to avoid unnecessary surgery [27]. The US Preventive Services Task Force, which reviews the effectiveness of screening programs in asymptomatic individuals, did not recommended screening for TC in adults without signs or symptoms of the disease [28]. The identification of thyroid malignancy is mainly based on ultrasonographic evaluation of suspicious TNs in patients reporting symptoms consisting of: globus sensation (sensation of a lump or foreign body in the throat); dysphagia or swallowing complaints (stasis, choking, odynophagia); dyspnea; dysphonia or hoarseness; and pain (due to acute increase in nodule size, as in case of bleeding into the nodule) followed by cytological examination of FNAB specimens [29]. The preoperative diagnostic performance of cytology, in terms of sensitivity, specificity, and accuracy in differentiating between benign and malignant lesions is of paramount importance. This task is particularly difficult in the group of cytologically indeterminate thyroid nodules (Bethesda classification III or IV), that pose a great challenge to pathologists, clinicians, and molecular biologists. In recent years, many papers and a lot of reviews have been published using this argument [30,31]. In all these reviews a detailed analysis of all the commercially available molecular tests available so far, based on the identification of several candidate genetic alterations associated with malignant transformation is reported. These molecular tests are proposed and currently used in the USA as well as in other HICs to correctly identify malignancy in cytologically indeterminate TNs [32]. Surprisingly, the possibility to use a validate cheap and reliable immunohistochemical approach for detecting TC preoperatively, especially in LMICs, is rarely reported and discussed.

## 5. Immunohistochemistry in Cancer Diagnosis

Immunohistochemistry (IHC) in cancer management was found to be extremely cost-effective in different clinical scenarios, both for diagnosis and prognosis [33]. Its indispensable role in cancer diagnosis on tissue specimens and/or cytological substrates has been well established for different cancer types [34]. The clinical utility of immunophenotypic analysis in the diagnosis of TC has also been clearly demonstrated, especially for Galectin-3 that is highly expressed in TC, but not in normal thyroid tissue [35] (Figure 1).

## 6. Advanced Molecular Tests for Thyroid Cancer and Their Cost

Many commercially available tests have been developed in recent years. These methods use advanced molecular analysis, in which the final diagnosis depends on a complex algorithm based on the results of a comprehensive panel of genetic alterations, including point mutations, rearrangements, gene translocations and gene fusions, as well as on the analysis of the expression of selected *miRNA* [36,37,38,39,40]. These tests are currently used in the USA and in HICs in Europe, where their costs are covered by insurance companies. The situation is different in LMICs where the costs should be covered by public health protection systems or by out-of-pocket payments. The European perspective on the use of molecular tests in TC diagnosis was not conclusive [41]. The utility of commercially available advanced molecular tests in reducing the number of patients in whom diagnostic surgical intervention is necessary was recognized, but the need of validation and prospective studies in European countries, particularly in LIMCs was stressed too, especially when considering the high cost of these tests (up to USD 3000) that may limit their availability. In this regard, also the NICE committee acknowledged that cost-effectiveness of advanced molecular testing was very dependent on the setting where the analysis was conducted [42]. In countries where TC management is expensive, like the US, advanced molecular testing is likely to be very cost effective, whereas in healthcare systems where it is cheaper, like Canada or the UK, its cost effectiveness is more uncertain. Moreover, in HICs, the costs for advanced in silico tests are usually covered by private insurance companies, whereas such costs are usually supported by general taxation and out-of-pocket payments in LMICs with public healthcare systems, responsible for an increase in the personal financial burden [43]. In this regard, it has been recently demonstrated that even in high-income countries, physicians would order fewer tests or search for lower-priced alternatives If hospitals showed the price of diagnostic laboratory tests at the time such tests are ordered [44].

## 7. Galectin-3

Galectin-3 belongs to the galectin superfamily, and it is the only member of this family to show a chimeric structure [45,46]. It is an interesting molecule, involved in many fundamental cellular processes, including cell–cell and cell–matrix interactions, growth, proliferation, differentiation, and inflammation and is involved in a multitude of tumoral as well as non-tumoral diseases [45]. Galectin-3 is ubiquitously expressed [47] and it is localized predominantly in the cytoplasm, but it has also been detected in the nucleus, on the cell surface, and in the extracellular environment. This molecule can interact with many other molecules, both inside and outside the cells, thus exhibiting specific pleiotropic biological functions. In the extracellular space, Galectin-3 modulates important interactions between epithelial cells and extracellular matrix and plays a fundamental role during the embryonic development of collecting ducts [48]. In contrast, intracellular Galectin-3 plays a relevant role in cell survival, because of its ability to block the intrinsic apoptotic pathway [49,50]. In the nucleus, Galectin-3 is localized mainly in interchromatin spaces, at the border of condensed chromatin, on the dense fibrillar component, and at the periphery of the fibrillar centers of nucleoli [51]. What makes this molecule of specific interest regarding cancer is represented by its antiapoptotic function [52,53] and its ability to form lattices with glycoproteins and glycolipids, implicated in regulating cell adhesion, metastasis, endocytosis, and other tumor-related biological processes [54,55]. For all these reasons Galectin-3 has been extensively studied in many different types of cancers, affecting various human tissues [45].

## 8. The Role of Galectin-3 in the Diagnosis of Thyroid Cancer

The role of Galectin-3 in cancers was analyzed in the thyroid. Almost 30 years ago, it was demonstrated that the detection of Galectin-3 expression, the cytoplasm of the thyroid cells showed a diagnostic hallmark of thyroid malignancy and was proposed as a presurgical diagnostic marker of TC [56], suitable to be applied to the cytological material obtained by FNAB [57]. The identification of Galectin-3 expression was especially effective in those lesions that were classified as indeterminate at cytological evaluation. Since the introduction of Galectin-3 based ThyroTest in the algorithm of suspicious thyroid nodules, we observed a dramatic reduction in the surgical procedure, manly performed to obtain a diagnostic characterization of the lesions. One of the major advantages of this test is the low cost (approximately 100 US dollars), and its accuracy has been validated in several multicentric clinical trials [58,59]. The guidelines for a correct Galectin-3 immunostaining procedure, applied to the cells retrieved by FNAB from suspicious TNs, have been described in detail [60]. Galectin-3 immunostaining allows to distinguish the thyroid carcinoma cells that appear colored in brown (Figure 2A) from the normal thyroid cells that are Galectin-3 negative at immunostaining (Figure 2B). Its accuracy can be further increased if another marker is analyzed in combination, namely the HMBE-1 [61], or with other markers of thyroid cell proliferation, such as cytokeratin-19 and the Ki67 index. The use of Galectin-3 as a marker of TC is also recognized in histology, where it is commonly used to confirm final diagnosis [62,63]. In addition, extended follow-ups of cytologically indeterminate TNs that were negative at Galectin-3 ThyroTest proved to be accurate, not only in identifying malignancy (rule-in ability), but also in excluding it (rule-out capacity) [64]. The relevant role of this test in the diagnosis has also been acknowledged by American Thyroid Association [65]. Considering the low cost of the Galectin-3 ThyroTest, its importance was recognized especially in LMICs [39,66]. Finally, Galectin-3 is one of the major characteristics of an ideal biomarker, because it is directly involved in the process that causes TC. On one hand, its forced expression in an in vitro model of normal thyroid cells (TAD-2 cells), via cDNA transfection, is capable to induce a transformed phenotype, consisting in the acquisition of serum-independent growth, clonogenicity in soft agar and loss of contact inhibition [67]. On the other hand, inhibition of Galectin-3 expression, by means of mRNA interference, in a TC cell line (NPA cells), was able to revert the transformed phenotype [68]. Moreover, useful insights regarding the mechanism of action of Galectin-3 have been reported. Galectin-3 is regulated at the transcriptional level by *wt* p53, and it plays a direct role in the mechanism of p53-induced apoptosis [69]. Gain-of-function p53 mutants, usually associated with the more aggressive forms of TC, are able to stimulate Galectin-3 gene expression, thus increasing chemoresistance [70]. Impairments of HIPK2, the master p53 regulator, frequently detected in well-differentiated thyroid carcinomas, are responsible for the occurrence of Galectin-3 overexpression observed in these types of cancer [71].

## 9. The Unappreciated Value of Galectin-3 in the Diagnosis of Thyroid Cancer

Economic modeling and related analytic techniques (cost–benefit, cost–effectiveness, estimated value of information) allows to assess the societal value of medical diagnostic testing [72]. Despite the reported evidence, the Galectin-3 ThyroTest is not considered and not even mentioned in many reviews focused on appropriate identification of TC [30,31,73,74,75,76,77,78], as well as in studies where the cost-effectiveness of the different approaches is comparatively evaluated [78,79,80,81,82,83]. In such studies, the cost is usually evaluated in comparison with surgical procedure and not with the low-cost methods, based on the use of IHC technique. In 2017, we performed a comparative analysis regarding the combined accuracy and costs, among different test-methods proposed to increase diagnostic accuracy in such lesions, including Galectin-3 ThyroTest, BRAF mutation analysis (BRAF), Gene Expression Classifier (GEC) alone and GEC + BRAF, mutation/fusion (M/F) panel, alone, M/F panel + *miRNA* GEC, and M/F panel by next generation sequencing (NGS), FDG-PET/CT, MIBI-Scan and TSHR mRNA blood assay [84]. We performed a systematic review and a meta-analysis to compare the feasibility of the tests, diagnostic performances as well as their costs. According to our analysis, the advanced molecular test-methods of GEC, GEC + BRAF, M/F panel + *miRNA* GEC and M/F panel by NGS were the best in ruling-out malignancy, with a sensitivity of 90%, 89%, 89%, and 90%, respectively. BRAF and M/F panel alone and by NGS were the best in ruling-in malignancy, with a specificity of 100%, 93%, and 93%, respectively. The M/F by NGS showed the highest accuracy of 92% and BRAF the highest diagnostic odds ratio (DOR) of 247. The advanced molecular tests require not only specialized instruments, but also highly trained staff to conduct the examination and then to interpret the reports. The entire process is therefore time consuming as well as expensive. Galectin-3 ThyroTest performed well as rule-out test with a sensitivity of 83%, as a rule-in test, with a specificity of 85%, and showed good accuracy of 84% and a high DOR of 27. In addition, it was one of the cheapest with USD 113, as compared to more than USD 3500 of the more complex molecular tests. Finally, it was the easiest one to be performed in different clinical settings. Such a comparative analysis demonstrated that this test could represent a cheap, and “good enough” method for the detection of TC, among the cytologically indeterminate TNs. Nevertheless, Galectin-3 ThyroTest has not gained the central role it deserves in the algorithm of suspicious TNs. There are many reasons why such a test is not preferred to the more expensive advanced molecular tests. Galectin-3 ThyroTest is an in vitro diagnostic (IVD) test, offered as a laboratory developed test (LDT), developed in hospital. It is not commercially distributed and is not protected by patent. As a hospital developed LDT, it has the advantage to be integrated into the continuum of patient care, and at the same time, it fulfills the many patient safeguards to which laboratories are already subject. In this regard, it should be considered that, for many years, the FDA has applied enforcement discretion to the development and use of LDTs in hospitals and health systems, to ensures that these clinical laboratory tests remain accessible, safe and effective. On the other side, the advanced molecular tests are commercially marketed IVDs, produced by private companies, and covered by patents. In the process of bringing advanced diagnostics to the market, companies have to provide evidence that the use of their diagnostic tests influences clinical interventions or improves patient outcomes more than existing tests. In other words, they have to prove actionability, before sustaining adoption of the tests. We believe that comparison with other available tests, which are “good enough” and much cheaper, would have affected the process of proving actionability. The recent expansion of these tests in high-income countries, however, has to face the bottlenecks of the reimbursement process by the insurance companies. Laboratories, in fact, may be caught in the middle between having to run them, at the request of providers and the possibility that they may not obtain reimbursment by insurers. The only way to demonstrate the clinical utility of low-cost non-commercially marketed IVDs is to perform accurate cost-effective analysis comparisons between such tests and the more expensive advanced molecular tests that we conducted (see reference # 84). The literature is full of such cost-effective analyses regarding this topic, but they do not even mention the low-cost methods. The advanced molecular tests are usually compared to surgery or to other molecular tests based on different methods. We believe that the issue deserves more attention, and we would like to offer to clinicians, that practice in all different economic conditions, the opportunity to better understand the advantages of this test and to consider its use in the management of suspicious thyroid nodules in the way we have performing for many years. We are confident that our present contribution, in a renowned scientific journal like JCM, will contribute to increase the prevalence of Galectin-3 ThyroTest in clinical applications.

## 10. Conclusions and Future Perspectives

Many efforts in recent years have been spent trying to make cancer tests easier and more affordable, and to catch cancer early and in as many people as possible, especially for the majority of cancers that are localized and can be cured by surgery alone, without any systemic therapy [85]. Ideally, these tests should be applied to the blood and should be able to detect cancers, as well as the tissue of origin, before the onset of distant metastasis. Among them, some tests have been focused on the combination of protein markers and genetic alterations using liquid biopsy [86]. Although these methods appear promising, the detection of malignant cells in blood in the early stages of disease may be difficult and it will need unbelievable sensitivity, which means unbelievable specificity in the binding reagents used [87]. Meanwhile, new, cheap, and sensitive methods will be available for the early diagnosis of TC, we may take advantage of the cheap and “good enough” IHC methods that we already have to reduce the socioeconomic burden of the management of the suspicious thyroid nodules, especially in those countries in which the total healthcare resources are finite.

TNs are rather frequent in the general population and represent an emerging global pandemic. The way we manage such a multitude of lesions has relevant economic consequences, especially if we consider that the vast majority of them are benign. The ideal test, chosen to identify the rare malignancy among the multitude of TNs, should fulfill the criteria of high accuracy, of low-cost and, possibly, of being pathogenetically relevant for the development of TC. Galectin-3 based ThyroTest represents an accurate test, with extremely low cost and is involved in the process of acquisition of the malignant phenotype of the thyroid cell. When compared the accuracy of this test with other advanced molecular test-methods, its performance is adequate enough to be included in the diagnostic flow chart of the suspicious TN. In the same diagnostic algorithm, we could consider the possibility of using advanced molecular methods as an adjunctive step, after the Galectin-3 ThyroTest, to better characterize those nodules in which the results of the Galectin-3 ThyroTest were not conclusive. Such a possibility could only be tested with the collaboration of the companies by means of large-scale clinical trials and bioinformatic analyses, in a long-term prospect.

Unfortunately, very few studies have performed comparative analysis between Galectin-3 ThyroTest and advanced molecular tests. For this reason, the utility of this test has not emerged in the way it deserves. Its use, therefore, is not as widespread as it could be. We believe that the lesson provided by the recent COVID-19 pandemic, that was responsible for a dramatic burden on healthcare systems, especially in LMICs, should be carefully considered [88]. The COVID-19 pandemic, in fact, brought home the value of cheap, “good enough” methods of detecting disease [89]. By extending the same approach to other illnesses and, in particular to the global pandemic of TNs, and by including Galectin-3 ThyroTest into the clinical management of cytologically indeterminate TNs, a significant improvement in healthcare especially, but not limited to LMICs, will be expected.

## Figures and Tables

**Figure 1 jcm-13-07290-f001:**
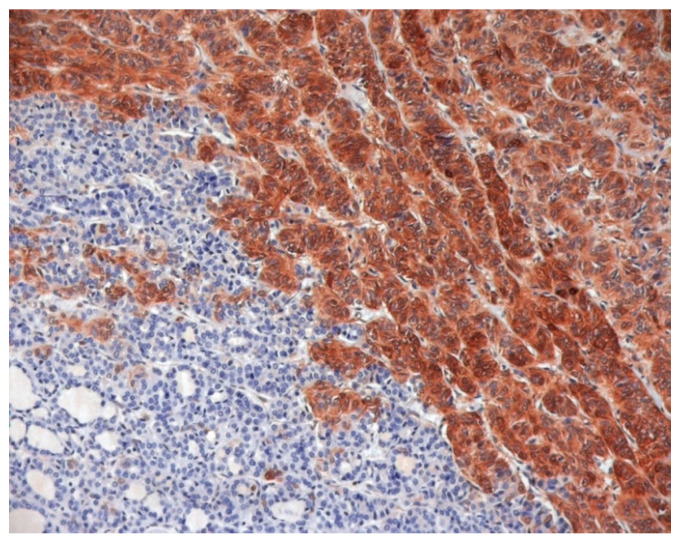
Galectin-3 Immunostaining of histological and cytological thyroid samples. A conventional formalin-fixed and paraffin-embedded histological section of a follicular variant of papillary thyroid carcinoma, showing galectin-3 positive TC cells and galectin-3 negative, adjacent, normal thyroid tissue.

**Figure 2 jcm-13-07290-f002:**
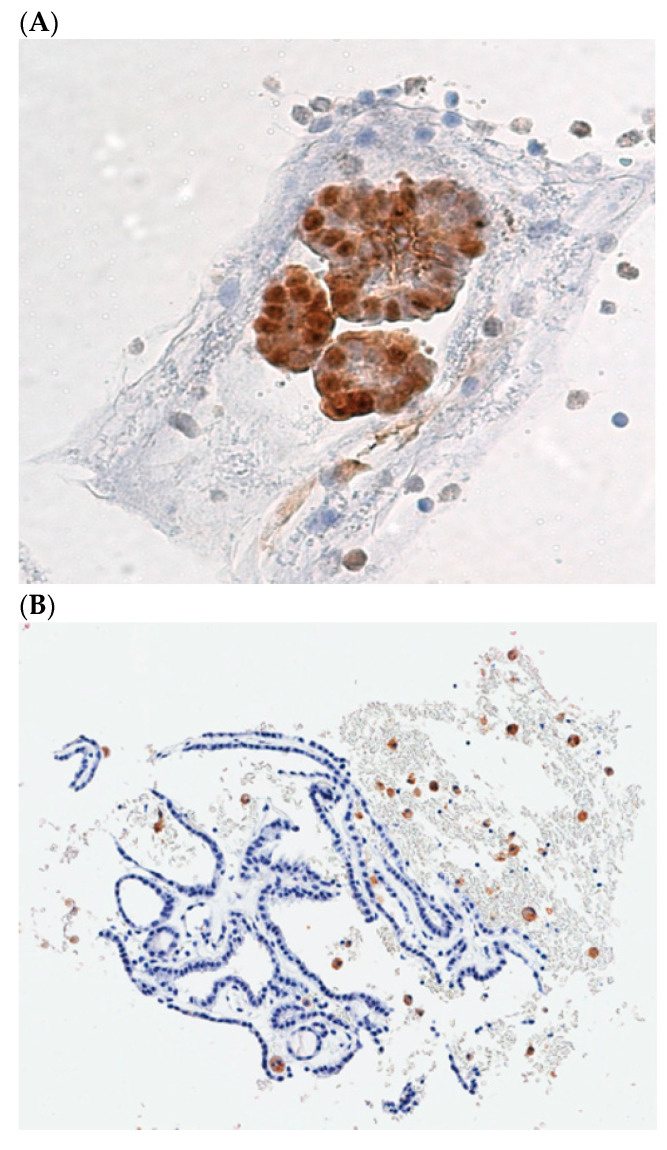
(**A**), Cellblock preparation of a FNA biopsy specimen showing galectin-3 positive TC cells. (**B**), A benign follicular thyroid proliferation with galectin-3 negative thyroid cells. Dispersed foamy macrophages, stained with galectin-3, are used as internal positive control.

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
