# Peer review of "The Unappreciated Value of a Cheap, ‘Good Enough’ Method of Detecting Thyroid Cancer"

_jcm, 2024, doi:10.3390/jcm13237290_

Round 1

Reviewer 1 Report

Comments and Suggestions for Authors

The authors realized that Galectin-3 ThyroTest performed with the IHC technique, which is a relatively reliable and cheap method compared to molecular genetic analysis in determining the malignant potential of thyroid nodules, which are frequently seen in society and constitute a significant burden on the health system, is not given enough importance in the clinical management of cytologically indeterminate TNs and conducted a review to draw attention to this issue. I would like to thank the authors for bringing this issue to the agenda. Indeed, in middle-income countries like the country where this reviewer lives, we cannot use molecular methods to determine cytologically indeterminate thyroid nodules due to cost. Unfortunately, most of the time, patients have to go to surgery for this reason. Because the cost of surgical methods is mostly based on the surgeon's labor (cheap manpower), which is less costly. Therefore, I appreciate that this issue has been brought to the agenda. However, the manuscript is generally well-designed, but there are sections that cause misunderstandings and are difficult to read due to typos. Just as an example, please pay attention to the first sentence in the introduction (in fact, the exact opposite of what is meant is what is written there). Therefore, I think that the entire paper should be revised from beginning to end in terms of language. Finally, the most important issue that I am curious about (although the authors briefly touched on it in the conclusion section) is why this method suggested by the authors not preferred by health systems and practitioners if it is as reliable and cheap as they claim. Why is the more expensive method preferred when there is such a cost-effective method? What are the shortcomings and limitations of Galectin-3 ThyroTest? Also, can you make suggestions regarding the clinical use of this test? For example, if there is still uncertainty after the application of Galectin-3 ThyroTest in thyroid nodules with uncertain cytology, can molecular methods be an option at the next stage? What can be done to increase the prevalence of Galectin-3 ThyroTest in clinical applications? These are my comments as far as I could review the manuscript. I wish the authors success.

Comments on the Quality of English Language

I mentioned in the comments above.

Author Response

Reviewer # 1

Comments and Suggestions for Authors

The authors realized that Galectin-3 ThyroTest performed with the IHC technique, which is a relatively reliable and cheap method compared to molecular genetic analysis in determining the malignant potential of thyroid nodules, which are frequently seen in society and constitute a significant burden on the health system, is not given enough importance in the clinical management of cytologically indeterminate TNs and conducted a review to draw attention to this issue.

I would like to thank the authors for bringing this issue to the agenda.

Indeed, in middle-income countries like the country where this reviewer lives, we cannot use molecular methods to determine cytologically indeterminate thyroid nodules due to cost. Unfortunately, most of the time, patients have to go to surgery for this reason. Because the cost of surgical methods is mostly based on the surgeon's labor (cheap manpower), which is less costly. Therefore, I appreciate that this issue has been brought to the agenda.

However, the manuscript is generally well-designed, but there are sections that cause misunderstandings and are difficult to read due to typos. Just as an example, please pay attention to the first sentence in the introduction (in fact, the exact opposite of what is meant is what is written there).

Therefore, I think that the entire paper should be revised from beginning to end in terms of language.

Finally, the most important issue that I am curious about (although the authors briefly touched on it in the conclusion section) is why this method suggested by the authors not preferred by health systems and practitioners if it is as reliable and cheap as they claim.

Why is the more expensive method preferred when there is such a cost-effective method?

What are the shortcomings and limitations of Galectin-3 ThyroTest?

Also, can you make suggestions regarding the clinical use of this test? For example, if there is still uncertainty after the application of Galectin-3 ThyroTest in thyroid nodules with uncertain cytology, can molecular methods be an option at the next stage?

What can be done to increase the prevalence of Galectin-3 ThyroTest in clinical applications?

These are my comments as far as I could review the manuscript. I wish the authors success.

Response to Reviewer # 1

First of all, we would like to thank Reviewer #1 for the useful comments and for the appreciation to our manuscript. We really think the issue is relevant and deserves consideration. As recognized by the reviewer, the high cost of the molecular methods to determine cytologically indeterminate thyroid nodules is the reason why such methods are not performed in middle-income Countries. The alternative is to direct patients with such nodules to surgery to obtain a diagnosis.

The entire manuscript has been revised in term of language according to the reviewer’s suggestion. We modified the first sentences in the Introduction to clarify better the concept.

We have added a comment regarding the reasons why the more expensive methods are preferred with respect to the Galectin-3 ThyroTest in high-income Countries. In general, our test should be considered as an in vitro diagnostic product (IVD), offered as a Laboratory Developed Test (LDT). Such test has been developed in hospital and is not covered by patent. The peculiarities of our test are the following:

- it has been developed, validated and performed in-house by our laboratory

- is not commercially distributed and is not protected by patent

- it is integrated into the continuum of patient care

- it fulfills the many patient safeguards to which laboratories are already subject.

On the contrary, the advanced molecular tests are commercially marketed IVDs, covered by patents, and produced by private companies.

FDA for many years has applied enforcement discretion to the development and use of LDTs in hospitals and health systems, to ensures that these clinical laboratory tests remain accessible, safe and effective. The commercially marketed advanced molecular tests have been pushed into the market by the companies in the last years, claiming that by reducing the cost of surgery, they are able to reduce the total costs of the management of suspicious thyroid nodules. However, the recent expansion of these tests in high-income countries has to face the bottlenecks of the reimbursement process by the insurance companies. As a consequence, laboratories are caught in the middle between having to run them, at the request of providers, with the possibility that they may not get reimbursed by insurers. The only way to demonstrate the clinical utility of low cost, non-commercially marketed IVDs, is to perform accurate cost-effective analysis comparisons between such tests and the more expensive advanced molecular tests as we did (see reference # 84). That is exactly the point. The literature is full of such cost-effective analyses regarding this topic, but they do not even mention the low-cost methods. The advanced molecular tests are usually compared to surgery or to other molecular tests based on different methods. We decided to write this manuscript because we believe that the issue deserves more attention and we would like to offer to clinicians, that practice in all different economic conditions, the opportunity to know better the advantages of this test and to consider its use in the management of suspicious thyroid nodules in the way we are doing since many years. We believe that our present contribution, in a renowned scientific journal like JCM, will contribute to increase the prevalence of Galectin-3 ThyroTest in clinical applications.

Of course, the possibility to use advanced molecular methods to better characterize those nodules that were analyzed using the Galectin-3 ThyroTest and where there could remain some doubts on the results of the test, is an option that could be tested. We would like to have the collaboration of the companies to perform such study. We reported these comments in the text of the manuscript. 

Reviewer 2 Report

Comments and Suggestions for Authors

Here, the authors discuss the cost-effectiveness and practicality of using Galectin-3 based immunohistochemistry (IHC) for the preoperative diagnosis of thyroid cancer, particularly in low- and middle-income countries (LMICs). The authors argue that while advanced molecular diagnostic techniques provide detailed insights into cancer biology, they are often prohibitively expensive and not accessible to all healthcare systems. The Galectin-3 ThyroTest, on the other hand, is a low-cost, effective method for diagnosing thyroid cancer in cytologically indeterminate thyroid nodules, making it a valuable tool in resource-limited settings.

Main points

The Galectin-3 ThyroTest is significantly cheaper than advanced molecular tests, making it accessible to a broader range of healthcare systems, especially in LMICs.

The test is easy to perform and does not require specialized equipment or highly trained personnel, which is advantageous in settings with limited resources.

The test has been shown to be effective in diagnosing thyroid cancer in indeterminate thyroid nodules, reducing the need for unnecessary surgeries and associated complications.

Points to adress

The study focuses primarily on the economic benefits and does not provide extensive clinical validation data, which may limit its acceptance in the broader medical community.

There is a lack of direct comparison with other low-cost diagnostic methods, which could provide a more comprehensive understanding of its relative effectiveness.

The study does not include bioinformatic validation of the findings, which could strengthen the evidence for the test's diagnostic accuracy and reliability, such as an integration of bioinformatics tools to analyze the molecular pathways and interactions involving Galectin-3.

Minor points. 

The authors should propose a validation of the test's accuracy and reliability through large-scale clinical trials and bioinformatic analyses, in a long term prospects paragraph.

Author Response

Reviewer # 2

Comments and Suggestions for Authors

Here, the authors discuss the cost-effectiveness and practicality of using Galectin-3 based immunohistochemistry (IHC) for the preoperative diagnosis of thyroid cancer, particularly in low- and middle-income countries (LMICs). The authors argue that while advanced molecular diagnostic techniques provide detailed insights into cancer biology, they are often prohibitively expensive and not accessible to all healthcare systems. The Galectin-3 ThyroTest, on the other hand, is a low-cost, effective method for diagnosing thyroid cancer in cytologically indeterminate thyroid nodules, making it a valuable tool in resource-limited settings.

Main points

  1. The Galectin-3 ThyroTest is significantly cheaper than advanced molecular tests, making it accessible to a broader range of healthcare systems, especially in LMICs.
  2. The test is easy to perform and does not require specialized equipment or highly trained personnel, which is advantageous in settings with limited resources.
  3. The test has been shown to be effective in diagnosing thyroid cancer in indeterminate thyroid nodules, reducing the need for unnecessary surgeries and associated complications.

Points to address

  1. The study focuses primarily on the economic benefits and does not provide extensive clinical validation data, which may limit its acceptance in the broader medical community.
  2. There is a lack of direct comparison with other low-cost diagnostic methods, which could provide a more comprehensive understanding of its relative effectiveness.
  3. The study does not include bioinformatic validation of the findings, which could strengthen the evidence for the test's diagnostic accuracy and reliability, such as an integration of bioinformatics tools to analyze the molecular pathways and interactions involving Galectin-3.

Minor points. 

  1. The authors should propose a validation of the test's accuracy and reliability through large-scale clinical trials and bioinformatic analyses, in a long-term prospects paragraph.

Response to Reviewer # 2

We thank the reviewer # 2 for the comments and suggestions.

  1. The clinical validation data have been published in many National and International multicentric clinical trials, published in the last 20 years (see the References #56, #57, #58, #59 and #64 in the manuscript). Moreover, the relevant role of this test in the diagnosis has been also acknowledged by American Thyroid Association (see reference 65).
  2. We agree that there is lack of direct comparison with other diagnostic methods. However, we think that the most relevant comparison should be not with other low-cost test method, but with the high-cost ones. We proposed to test the accuracy of our method and of the advanced molecular diagnostic tests in the same population of suspicious thyroid nodules, but we didn’t find enthusiasm to our proposal and we couldn’t perform such comparison without the collaboration of the companies that own the property of tests.
  3. The Galectin-3 ThyroTest is based on the recognition of staining into the cytoplasm of the thyrocyte, which can be easily detected by the pathologist and, eventually, reported in grades based on the intensity of staining. There is no need of bioinformatics to obtain the results and this is the reason why is so cheap and easy to be performed. We reported the experimental studies concerning the interaction of Galectin-3, performed by our group as well as by other groups, in which the molecular pathways and interactions involving Galectin-3 have been analyzed (see References #67, #68. #69, #70, and #71).

4. According to the Reviewer’s suggestion, we added a sentence in the last paragraph to propose validation of the test's accuracy and reliability of our ThyroTest in comparison with other more expensive tests, through large-scale clinical trials and bioinformatic analyses, in a long-term prospect.  

Reviewer 3 Report

Comments and Suggestions for Authors

Thank you for the opportunity to review of manuscript entitled "The unappreciated value of a cheap, ‘good enough’ method of detecting thyroid cancer".

I read the manuscript with great attention - I believe that the topic is worthy of it. I congratulate the authors for undertaking this task.

The abstract is coherent and adheres to the standards of proper composition; yet, its excessive length (about 341 words) will likely require compression.

The introduction succinctly acquaints the reader with the subsequent sections of the manuscript. I have no reservations regarding this piece.

The division of the manuscript into smaller theoretical sections is clear and effectively guides the reader from general to specific regarding the topic of thyroid cancer diagnostics with galectin-3.

Nevertheless, I think that the article lacks a comparison of the use of galectin-3 with other potential markers of thyroid cell proliferation (such as cytokeratin-19) and a link to the Ki67 index. Although I do not consider it a mandatory element of this manuscript, I encourage the authors to consider adding a few sentences or paragraphs on this topic - it will certainly increase the substantive quality of the article.

Despite the above, I congratulate the authors on taking on this topic and wish them further success in their scientific careers.

Author Response

Reviewer # 3

Comments and Suggestions for Authors

Thank you for the opportunity to review of manuscript entitled "The unappreciated value of a cheap, ‘good enough’ method of detecting thyroid cancer".

  1. I read the manuscript with great attention - I believe that the topic is worthy of it. I congratulate the authors for undertaking this task.
  2. The abstract is coherent and adheres to the standards of proper composition; yet, its excessive length (about 341 words) will likely require compression.
  3. The introduction succinctly acquaints the reader with the subsequent sections of the manuscript. I have no reservations regarding this piece.
  4. The division of the manuscript into smaller theoretical sections is clear and effectively guides the reader from general to specific regarding the topic of thyroid cancer diagnostics with galectin-3.
  5. Nevertheless, I think that the article lacks a comparison of the use of galectin-3 with other potential markers of thyroid cell proliferation (such as cytokeratin-19) and a link to the Ki67 index. Although I do not consider it a mandatory element of this manuscript, I encourage the authors to consider adding a few sentences or paragraphs on this topic - it will certainly increase the substantive quality of the article.

Despite the above, I congratulate the authors on taking on this topic and wish them further success in their scientific careers.

Response to Reviewer # 3

  1. We would like to thank the reviewer 3 for his/her comments and for wishing us success in our scientific careers. We really appreciated the congratulations for our manuscript.
  2. We have reduced the length of the Abstract
  3. We thank the reviewer for the appreciation of the Introduction
  4. We thank the reviewer for the appreciation of the division of the manuscript into smaller theoretical sections.
  5. We have added few sentences regarding the utility of other protein markers, such as cytokeratin-19 and Ki67 index, in section #8. However, in our experience, we couldn’t ameliorate the accuracy of Galectin-3 ThyroTest by adding these markers.